# An Exploration of the Mechanism of Action of an Equine-Assisted Intervention

**DOI:** 10.3390/ani9060303

**Published:** 2019-05-31

**Authors:** Ann Hemingway, Sid Carter, Andrew Callaway, Emma Kavanagh, Shelley Ellis

**Affiliations:** 1Department of Medical Sciences & Public Health, Bournemouth University, Poole BH12 5BB, UK; 2Department of Social Sciences & Social Work, Bournemouth University, Poole BH12 5BB, UK; scarter@bournemouth.ac.uk; 3Department of Sports & Physical Activity, Bournemouth University, Poole BH12 5BB, UK; acallaway@bournemouth.ac.uk (A.C.); ekavanagh@bournemouth.ac.uk (E.K.); sellis@bournemouth.ac.uk (S.E.)

**Keywords:** equine assisted, mechanism of action

## Abstract

**Simple Summary:**

Although there is increasing international interest and a growing body of evidence discussing the potential impacts of equine-assisted interventions to assist those with behavioural, mental health, physical health and disability-related issues, there has been little exploration of what “the mechanism of action” may be in causing any potential positive impacts. This paper reports on multi-method research study which considered in detail what was occurring while participants undertook the equine-assisted intervention under study. The intervention was implemented with young people with chronic mental health and behavioural problems for whom talk-based interventions were not working. Previous research has demonstrated long-term health and wellbeing benefits in recipients of the intervention. The three datasets were video data, psycho-physiological data, in this case, skin conductivity response, which is an indicator of emotional arousal, and experiential interview data. Our findings indicated that learning natural horsemanship skills through this intervention caused participants to experience emotional arousal when they asked the horse to perform a task. We would suggest that this process of experiencing a positive outcome following emotional arousal helps participants to achieve the reported behavioural outcomes from this intervention which include increased calmness, assertiveness, focus, empathy, communication skills, taking responsibility for behaviour, planning and confidence as a learner. These changes in behaviour then translate from the intervention to the participants’ everyday lives, thereby achieving the changes in behaviour recorded by those independent practitioners (i.e., social workers and teachers) who refer individuals to this intervention.

**Abstract:**

Though long alluded to, there is now an accumulation of evidence of the vital contribution that emotion makes to learning. Within this broad advance in understanding is a growing body of research emphasising the embodied nature of this emotion-based learning. The study presented here is a pilot study using a mixed-method approach (combining both physiological and experiential methodologies) to give a picture of the “emotional landscape” of people’s learning through the intervention under study. This has allowed researchers to examine mediating pathways that may underlie any effects of an equine-assisted intervention. This study specifically focuses on examining the role of emotion. The intervention under study was used with young people with chronic mental health and behavioural problems for whom talk-based interventions were not working. Nine healthy participants aged 18–24 undertook the equine intervention, with an initial group having emotion-related psycho-physiological changes (skin conductance responses) measured while viewing their experience on video, and a further two participants experiencing a development of the methodology as their physiological responses were captured in real time during the intervention. The sessions were analysed by a group of five cross-disciplinary researchers to determine when significant learning episodes occurred, and the findings were that this learning was associated with powerful skin conductance responses. The qualitative element of the research entailed the participants watching themselves on video undertaking the equine intervention. They were asked to stop the video and share any changes in emotion at any point while watching. All participants experienced a positive temporal change in mood as the intervention progressed. All results supported the findings that emotional arousal occurred in relation to the participants asking the horse to perform a task. This paper will offer two novel contributions: (1) description of a new methodology for investigating the mechanism of action occurring in this type of intervention and (2) findings from the exploration of the intervention via psycho-physiological and experiential mechanisms.

## 1. Introduction

There is now growing evidence of the contribution that emotion makes to learning. A growing body of research emphasises the embodied nature of this learning. The present study is a pilot study (combining both physiological and experiential methodologies) to give a picture of the “emotional landscape” of people’s learning through the intervention under study. This has allowed the researchers to explore the possibility that emotions may contribute to learning which occurs through equine-assisted activities.

Despite its growth in popularity globally, little is known currently about the mechanism of action of inter-species equine-assisted interventions (EAIs) [1,2]. In the broader equine intervention related literature, there have been individual studies on equine-assisted therapy with individuals with disabilities [3,4,5,6,7,8,9].

Studies have also been published on the impact of equine-assisted interventions on those living with chronic illness [10,11,12,13] or eating disorders [14]. In addition, the potential benefits of equine-assisted psychotherapy and counselling have been studied, although the outcomes have been mixed with some studies showing potential positive results [15,16,17,18,19] and some no effect [20].

Qualitative research has suggested emotions that may be experienced through interacting with horses and these include trust, motivation, patience respect and empathy [21,22]. In addition, some studies suggest that such interactions may reduce anxiety and help develop mental and emotional control [23]. Research with young vulnerable adults has found reductions in disruptive behaviour and improvements in relationships with others and self-esteem [24]. Research by Dell [25] captured enhanced communication skills and a sense of pride which the young participants experienced through communicating with horses. Hemingway et al. [2] gained insights into the positive impacts of an EAI with prisoners in a young offender’s institution, with the young men reporting that they felt calmer and more positive about their other educational opportunities. Prison guards reported marked improvements in behaviour in the prison overall and a reduction in one-year reoffending rates. 

Pendry and Roeter [26] undertook a randomised controlled trial to evaluate the effectiveness of an EAI on improving child social competence. Their findings reported some moderately significant improvements in social competence for 5th to 8th grade youngsters after the intervention. Further research is required to capture insights other than the participant’s parents into potential improvement. 

In a systematic reviews on animal-assisted interventions including equine interventions [27], the authors commented that the published studies examining the effectiveness of interventions involving horses lack specific details of the horse’s role, and the type of horsemanship development offered to the participants.

Two systematic reviews [28,29] have been published on the impact of equine-assisted therapy on physical symptoms in adults living with autism and schizophrenia. All reviews found that studies lacked rigour, randomisation, and adequate sample size. The autism review also commented on the variety of interventions offered and the lack of clarity in relation to what was included in the intervention with large varieties in experience for participants.

An umbrella review of equine-assisted interventions was published early in 2019 [30], which included any interventions that involved a live horse, for humans of any age, and for any therapeutic purpose. This review suggested that the evidence underpinning equine-assisted interventions was equivocal, and as such, it cannot be recommended as a best practice. However, the intervention under study here is being used extensively in practice and those referring individuals are consistently doing so for participants for whom talking therapies are not working (500 participants so far), and the same intervention has been rolled out across four different counties in the south of England. The intervention (EAI) is educational and teaches introductory horsemanship skills with the participant on the ground and the horse wearing a halter being managed by the human on a twelve-foot rope.

A review of animal-assisted therapy (AAT) for young people published in 2016 [31] included papers involving horses, dogs, cats, and dolphins and stated that the area itself showed promise in terms of therapeutic effects. The review, however, found that research quality was variable in this area and more focus is needed on research design to include control groups and random allocation of participants. In addition, the review suggested that further study on how interventions work (mediation) and with whom (moderators) is also required. Interestingly, it was intimated in the paper that research has not yet explored whether the mere presence of an animal causes any therapeutic effect from the interventions studied.

Overall, research to date has found that the interaction between humans and animals including horses may have an impact on human mental health and wellbeing and/or behaviour. However, the quality of the evidence is poor, with mainly non-randomized designs being published and evidence of a causal link between the interaction with the horse and human being evidenced clearly in only a small number of studies [30]. In addition, evidence thus far has not always clearly defined the nature of the intervention under study, the details of whether the intervention is based on equine-assisted learning, equine-assisted psychotherapy, therapeutic riding, and hippotherapy or caring for horses, for instance, is not always clear in the published article. The literature seems to be rich in systematic reviews but lacking in randomised controlled trials with clearly outlined intervention processes and clearly defined intervention groups. Indeed, all these different types of equine-assisted activities are in themselves not currently clearly defined. The largest organization globally is PATH International, which stands for the Professional Association of Therapeutic Horsemanship. However, the intervention under study here defines itself as an educational intervention assisted by horses and not a “therapy”. The participants learn level 1 horsemanship skills over a ten-hour period with horsemanship instructors not therapists. 

Although this area is not well studied in a traditional sense, such as may be undertaken with other health-related interventions, the level of interest and focus on it internationally and the numbers of individuals being referred to equine assisted activities has led to attempts to understand what mechanism may be at work when humans interact with horses. This literature has also been contributed to by those interested in how humans and horses learn and communicate together [32].

In an attempt to further understand the mechanism of action in EAIs, one pilot study was undertaken looking at the effects of EAIs and therapy on resting brain state function in attention-deficit/hyperactivity disorder [33]. This study found that the intervention was associated with short-range functional connectivity in the regions of the brain related to the behavioural inhibition system, which are associated with symptom improvement. In addition, a recent pilot study [34] also reported on heart rate, heart rate variability, and salivary cortisol measurements in both humans and horses and reported reductions in cortisol for human participants post-intervention.

## 2. Emotions and Learning

A growing body of literature suggests that somatic states related to emotion are involved in cognitive processes, including learning [35]. An example of this development is Damasio’s somatic marker hypothesis (SMH) [36,37], which states that for every life event, somatic consequences are marked and then reproduced when that event recurs. This assists in making decisions about the benefits of one particular option over others in terms of behaviour and enables us to make rapid decisions based on previous experience. These somatic consequences are strongly connected with the emotion systems of the brain (and the body) 

Further support for the link between emotions and learning, however, has come from various sources [38,39], Critchley et al. [40] found that using functional magnetic resonance imagery that changes the activity of the prefrontal cortex were related to spontaneous fluctuations in skin conductivity responses (SCR, measuring attention/cognitive effort/arousal). Since then, similar effects have been found using positron emission tomography [41]. In 2004, research was undertaken with healthy volunteers to examine whether a stronger autonomic response accompanies better learning, essentially testing out the link between emotions and learning [38]. The research found that positive emotions can enable effective learning and behavioural change.

Falk and Gillespie [42] published the relationship between emotional arousal and greater short- and long-term learning outcomes at a science centre exhibition about fear, in which visitors to the fear exhibited higher levels of emotional arousal than other visitors and demonstrated greater learning outcomes both immediately after their visit and 4–6 months later than control individuals who described an exhibit of their choice. In addition, Stauss [43] found significant positive correlations between aquarium visitors’ arousal levels and their ability to describe exhibits 2–3 months after their visit. Smith et al. [44], however, found a negative correlation between emotional arousal and learning during a birds-of-prey animal demonstration at a zoo, as those attending were focused on the birds flying near them and were, thus, not able to focus on and learn the associated conservation messages. The challenge, therefore, may be to manage the threshold between experiences that are exciting enough to elicit attention and learning, yet not to provoke anxiety so that learning does not occur. In 2017, Stauss and Faulk [45] examined the role of emotion in the context of a science learning experience to investigate the relationships between emotional arousal, valence, attention, environmental values, and learning outcomes. Sixty undergraduate and graduate students participated in one of two treatments consisting of watching an exciting or neutral nature documentary video, reading an associated narrative, and taking a post-test. The findings from that study suggested that higher emotional arousal led to greater short-term learning outcomes. However, in relation specifically to embodied learning, such as in the study presented here, these areas have still not been examined.

The relationship between emotions and learning is therefore well established in theory [38,46], and some evidence has been published as presented above; however, this relationship has not been tested in real-world applications specifically focused on behavioural change. 

The study reported in this paper was designed in order to explore the potential relationship between emotions and learning in the context of an EAI and to develop research methodologies in order to explore and describe this process further. This paper presents the findings from our study in order to attempt to understand whether interventions which include horses may be tapping into this emotional learning mechanism. 

## 3. Material and Methods

The intervention under study here used the principles of the Parelli Natural Horsemanship program as its philosophical basis and structure [47]. The principles involve action-based, experiential learning based on cooperation and partnership development between horse and human. At the introductory level, this involves “playing” with horses and inviting them to respond to requests with the person working from the ground with the horse on a loose rope. All the learning occurs in an indoor riding “school” with a special soft surface. The learning of each participant was facilitated by a course facilitator (a licensed Parelli horsemanship instructor who has also undertaken additional training specifically for this educational program) and they were taught how to play seven “games” [47] with the horses. The games taught are:(1)*The friendly game* (creating relaxation through touch, grazing, grooming, hanging out).(2)*The porcupine game* (moving the horse’s feet, touching the horse).(3)*The driving game* (moving the horse’s feet through rhythmic action, not touching the horse).(4)*The yo-yo game* (moving the horse backwards and forwards).(5)*The circling game* (asking the horse to travel around you on the circle).(6)*The sideways game* (asking the horse to move sideways).(7)*The squeeze game* (asking the horse to go through, under or over something).

This EAI uses a clear model to teach participants these seven games; the participant is set a task with the horse and is coached to success, improving their skills. They are then given time to celebrate success and this is repeated through moving to another task or skill. The lasting skills taught which have emerged from evaluation of the intervention under study here [2] to date are: calmness, assertiveness; holding a strong focus; analysing and planning; observing the needs of others/empathy; communicating clearly; engaging in new learning; taking responsibility for one’s own thoughts, feelings and actions. 

All young people involved in this pilot study were university students undertaking a sports science degree or a sociology degree in a university in the South of England. Three boys and six girls all aged between 19 and 24 volunteered to undertake a shortened half hour version of the EAI, essentially learning all seven games as outlined above. These are the skills normally taught to participants on the EAI but in a shorter period of time, and all were videoed while they undertook the course. Normally, participants spend ten hours on the course broken down into five two-hour sessions, where they learn these same seven games more slowly and accounting for their own individual needs as a learner.

These games help to establish a simple communication between horses and humans. In order to be effective, the human needs to use clear, phased assertive communication through their body language and control their emotions in an assertive, non-aggressive way. The course teaches through simulation and uses non-verbal methods (through rehearsal). The charity provides this program to 150–200 people each year, referred by schools and pupil referral units, children’s services, mental health services, offender services, and other specialist agencies (such as charities working with domestic violence or drug and alcohol services). Currently, the charity is based in the counties of Dorset, Gloucester, Kent, and London. The course participants have typically more than four issues from those included here.
attention deficit hyperactivity disorder,autism spectrum disorder, conduct disorder,anxiety,not attending school/training/work,relationship difficulties,mood swings/impulsivity,self-harming/being bullied,bullying, aggression/anger management issues,risk taking behaviour,drug and alcohol misuse,eating disorder,offending,domestic violence,witnessing domestic violence,neglect/abuse,poor parenting,parents with mental health problems,offending or drug and alcohol issues,living in care or leaving care.

The participants are referred because they are “stuck” or disengaged from talk-based support and education and are mainly referred by social services or Children and Adolescent Mental Health Services (CAMHS).

## 4. Ethics Approval

Ethics approval was obtained from the Ethics Committee of Bournemouth University. Written informed consent was obtained from all participants (BU REF ID 339). All the responses from the participants were presented without identifying the individuals. The video stills of the participants had the faces pixilated out to prevent identification or were video stills from the rear view of the individual. This registered charity undertakes risk assessments for all participants who are never left unsupervised with the horses. The horses are all observed for possible stress/distress continuously throughout the course activities, which would be ceased immediately if any observations of this were made [48]. This study recruited “healthy participants” to be involved in this initial pilot study, as exposing vulnerable young people normally referred to this intervention to physical measurement, etc., when the research team did not know if any relevant findings may be gained seemed ethically inappropriate at this stage in developing insights into this area.

## 5. Hetero Phenomenology

First person data are not about objective functioning, they inevitably relate to our beliefs and understandings of what is happening to us, as Dennett suggests we have an inkling of [49,50,51]. Psycho-physiological measures, however, may offer insights into objective functioning and in relation to SCR into emotional arousal, and therefore, objective measurable embodied experience.

Hetero-phenomenology has been described as spanning a chasm between the subjectivity of human experience and the natural sciences [49]. This study by attempting to start to understand a therapeutic phenomenon (mechanism of action), which has been studied to some extent on the phenomenological side of the chasm and very little from the natural science side, has tried to gather data from as many perspectives as possible to help to illuminate what may be happening to produce any effect. In addition, the use of this methodology has enabled researchers to consider embodied cognition “in real time” as it were, to help understand what can be considered “cognitive”. This study did not consider the mind as an independent entity or “disembodied”, but as a part of the body, indeed, we would suggest that the body and the embodied emotional experiences of our participants influenced their minds and their mood rather than the other way around [50]. Indeed, it has been suggested [51,52] that mental representations rather than embodied responses are not the best theoretical option for explaining human cognition.

## 6. Subjective Reports and Objective Behaviour

We utilised a methodology which is based on a rethinking of the standard cognitive “mapping” paradigm. This methodology allows psycho-physiological measurement to be triangulated with directly observed behaviour and introspective evidence [53]. The triangulation occurred by synchronizing the data in time in order to show at any given moment in the intervention what was occurring for participants in reality through the video data, emotionally through the psycho-physiological data, and at that point in time following the intervention, when they observed themselves on video in the phenomenological interviews. The reason for taking this path was that, through our interdisciplinary work, it became clear that our embodied and cognitive learning functions in real time with horses may be better understood with a broader perspective focused on the “dynamic story” rather than a pure focus only on stimulus and response [54]. Obviously, much more work needs to be done in order to understand how these different layers of “description and observation” work together.

Another way of using this mixture of data collection has been utilised in neuro phenomenology [55,56,57], where first person accounts are collected about moment-to-moment subjective experience while an individual views images and has their brain function recorded (fMRI) in order to offer insights into dynamic brain activity at that moment. The assumption underpinning this methodology is that these datasets can stand in a mutually constraining and illuminating relationship about the psychophysiological processes underpinning moment-to-moment experience [58]. How is our body reacting to what is happening to us at that moment in time? While the video data of what is happening in real time with the participant and the horse helps us to understand when and how this experience is impacting on our psycho-physiological embodied state in that moment.

Consider that at any moment our engagement with our environment is complex and filled with experiences such as sight, smells, tastes, touch, and more. In addition, our experiences are impacted on by our current state of attention, motivation and emotion [59]. Despite all of this complexity, however, each perceptual moment in time is experienced by us in a unified way, as a whole moment [60]. This research study has attempted to collect data on as many aspects of this “whole” as possible within the limitations of the practical nature of the intervention under study and the current limits of technology in terms of capturing data. 

Through capturing the individual’s internal “landscape” at one point in time (through measuring psycho-physiological data and recording first person accounts) and placing it in their external “landscape” (through observing their behaviour and the horse’s behaviour on video), we may begin to understand the layers of experience and learning occurring at one point in time.

## 7. Methods

The following methods of data collection were used to inform this study:Skin conductivity response (SCR).Qualitative interviews undertaken while participants watch themselves on video with the horses to help understand their emotions during this process.Observation of the individual and the horse’s interaction on video synchronized with the SCR data and the qualitative responses.

It might appear more logical to record psycho-physiological responses “live” during the activity itself. This was attempted during our initial pilot in February 2016 but was unsatisfactory for several reasons. Firstly, psychophysiological measures are easily confounded by movement artefacts, and the intervention under study is a physically active learning experience. Therefore, recording was unreliable, and good skin contact with electrodes was difficult to maintain. The EAI also requires extensive use of both hands, which in practise precluded the use of the SCR measure, which, until recently, has been most reliably captured using electrodes on the fingers. However, it was found during this pilot that watching a video of the procedure within 24 h after the event evinced strong emotional responses. This indicated that participants could “relive” the experience, but the psychophysiological recordings could take place in stable conditions. There is evidence that watching yourself on video undertaking a task is able to evoke embodied emotional recognition of the quality of the lived experience [61], which the research team further tested in this study. However, the technology became available to record SCR through a device attached to the wrist during our study (2017), so we tested the use of this device on the final two participants in real time while they undertook the EAI. We found that the SCR responses were still clear and consistent and occurring at the same points in the experience for participants, and that putting the video data with the SCR we could ensure that movement artefact was not mistaken for SCR response.

All young people involved in this pilot were university students undertaking a sports science degree or a sociology degree in a university in the South of England. Three boys and six girls all aged between 19 and 24 undertook a shortened half hour version of the EAI essentially learning the skills taught to participants normally referred but in a shorter period of time, and all were videoed while they undertook the course. Normally, participants spend ten hours on the course broken down into five two-hour sessions.

## 8. Phenomenological Interviews

Phenomenology is a qualitative research approach that concentrates on the study of consciousness and the objects of direct experience. It seeks to understand how people experience a particular situation or phenomenon and can be defined as the direct investigation and description of phenomena as consciously experienced by people living those experiences [62]. Phenomenology is centred on the participants’ experiences with no regard to social or cultural norms, traditions, or preconceived ideas about the experience. As a research methodology, it is based on the academic disciplines of philosophy and psychology and has become a widely accepted method for describing human experiences. A phenomenological study attempts to set aside biases and preconceived assumptions about human experiences, feelings, and responses to a particular situation. 

Phenomenology was used in this study in order to enable the researchers to delve into the perceptions, perspectives, understandings, and feelings of those people who have actually experienced or lived the phenomenon of learning to communicate with horses. Phenomenological research is typically conducted through the use of in-depth interviews of small samples of participants. By studying the perspectives of different participants, the researchers can begin to generate themes regarding what it is like to experience the phenomenon from the perspective of those that have lived the experience [62]. Phenomenological analysis includes an attempt to identify themes or make generalizations regarding how a particular phenomenon is actually perceived or experienced.

In this study, the interview prompts were open-ended to allow the participants to fully describe the experience from their own view point. The only prompts used were to ask the participants to stop the video if they felt anything in relation to the experience with the horses and describe their feelings or emotions; these could be “felt” in their body or in their mind or thoughts. The interviews in this study were undertaken while the participants watched themselves on video interacting with the horses. Participants were informed they were free to pause the video at any time and were asked to share their experience of any emotions or feelings they had during this period, no other prompts were used, and the words used by the participants were not prompted or suggested by the researchers in any way. The words emotions, feelings, and sensations were used in an attempt to enable and encourage the participants to verbalise any changes for them in terms of their “felt” experience while acknowledging that these remain contested terms to some extent in the academic literature [61]. It was felt that the widest use of terms would help the participants to feel free to explore any “felt” changes.

Two of the researchers analysed the findings from the interview transcripts independently and then jointly generated themes through careful word-by-word, line-by-line analysis of the responses, looking for similarities and differences in the participant responses, identifying relevant quotes, and preparing the findings section presented here. All phenomenological analysis involves an element of reflection, and differing approaches value either the use of organised and systematic processes or the emergence of slowly developing insights or “phenomenological sensibilities” in relation to what the participants’ lived experiences are offering. In the case of this study, the analytical method suggested by Wertz [62] and Giorgi [63], including systematic readings of the transcript, was undertaken by first dwelling on the participants’ descriptions of the phenomenon (through immersion and reflection), then description of the emergent constituents and recurrent themes was undertaken independently. Then, following this undertaking, the two researchers, in contrast, with dialogical analysis, used open, spontaneous, dialogue together to agree on the final emergent themes presented here. 

## 9. Interview Findings

### 9.1. Temporal Shift in Emotions during this Learning Experience

The participants all recounted the experience of a temporal shift in the emotions they experienced over their time interacting with the horse (30–35 min). Here, we present the themes which emerged from the participants’ transcribed interview data, the interviews were undertaken with the first seven participants while they watched themselves on video learning to play with the horses; for the final two participants, these SCR responses were recorded in real time while they played with the horses as described above. In addition, for these final two participants, interviews were then done while they watched themselves on video playing with the horses.

### 9.2. Initial Feelings of Fear

Six participants recounted feelings of nervousness, uncertainty, anxiety or fear at the start of the experience. These initial emotions were linked by the participants to the unknown nature of what was to happen and their potential interaction with the horse.
“I was very nervous cos I had not even considered how they might be scared of me. I did not really want to be in there. Yeah, I did not really think about it.”(Participant, P 2)
“When we walked into the arena, I was very nervous. I was aware I was going to like approach two like massive horses…that was fear…yes, well I worked on a beach and I saw horses like bucking and throwing their owners off and stuff. I am not sure about how they will behave and stuff like that.”(Participant, P 3)
“When I was further away from the horses, I was much more relaxed.”(Participant, P 4)
“I was quite nervous as I had never really had any contact with horses before.”(Participant, P 5)
“Yeah, a bit anxious I guess…to get started.”(Participant, P 6)
“Just not sure really.”(Participant, P 7)

Inevitably, the presence of a large charismatic mammal is likely to induce some feelings of nervousness, especially when one is aware that you are just about to be taught how to communicate with that mammal. It is interesting to consider, however, that these initial feelings may be part of the mechanism of action in that they may help to focus the attention of participants with behavioural and mental health issues and help with feelings of achievement and success, as one is coached to learn how to communicate with them effectively and safely. In reality, however, this needs further study in order to clarify whether the level of success of the intervention is in any way linked to the level of fear/nervousness at the start.

## 10. Confusion

The participants undertook a series of tasks which were introduced and explained by the facilitator; these demonstrated all of the Parelli seven games as outlined here previously. When any new task was introduced all participants referred to confusion initially due to the lack of understanding of how the communication with the horse would work to enable them to complete it. At this introductory level, the skills being taught involve “playing” with the horse and inviting them to respond to requests on a long rope with participants on the ground. The confusion articulated by six of the participants appeared to be a result of a lack of understanding of how the communication with the horse would work to enable them to complete the task with the horse.
“It felt confused and higgledy piggledy.”(P 3)
“I felt awkward, yes confused.”(P 6)
“I was really confused at this point…I thought why is she (the facilitator demonstration of leading the horse) not talking to the horse, and then I realised that she is just like picking herself up and that was going to make the horse realise that you were just about to go. I was very surprised that it would take so little.”(P 2)
“I just didn’t know how it was going to work.”(P 1)
“I got quite confused cos I did not know what that was at all (wiggling the rope to ask the horse to back up).”(P 2)
“I was like aren’t we going to talk…even though the horse does not speak ‘human’, I thought…like the way I talk to my dog though half the time it does not understand me…I was still expecting to speak and then the horse would recognise the tone of my voice and like respond to that, it just confused me.”(P 5)

The participants’ responses clearly indicated that they did not understand how they could communicate with a horse and what mechanism might be at work to enable this to happen at this point in the intervention.

## 11. Working out What Was Happening

At this stage, six of the participants described a strong sense of confusion due to the challenging nature of the learning experience where verbal communication with the horse is not available, and they were asked to work on a more embodied level. Participants then expressed a desire to work out what was happening, they felt puzzled and tried to put into words how the communication was working.
“I started a bit like suspicious like the horse knows what I am doing but at that point…I realised it was literally your energy if you stood up tall, he would follow you. It was literally like relax, he was not going to follow you...I did not expect it.”(P 4 )
“All the time I was trying to figure out how it works just through wanting something with your energy, not really doing much of anything you can get it…astonishing.”(P 1)
“I really did feel that maybe he’s responding to my energy. I don’t really know what that means…it’s like overwhelming.”(P 7)
“Yeah all I had to do was run…yeah it was strange.”(P 1)
“How would you do it, she said, like move the horse around the cone. I was like I don’t have a clue how to do it.”(P 2)

The ability of the horse to understand and respond to body language gradually became clear to the participants as the intervention progressed. They expressed how sensitive they were and how strange and different it felt to communicate consciously through their bodies.

## 12. Feeling Proud

The emotional experience of participants appeared to be directly linked to perceived task success, they did not want to “fail” or “get it wrong” (P 1) and six participants expressed very positive emotions when able to undertake a task successfully with the horse.
“I think with every little thing that you do right, it builds your confidence more, you gain a new skill.”(P 4)
“Makes you feel like you have got a lot of power, strong and powerful.”(P 5)
“When I got it right, when I got it properly right, I was very proud of myself, very proud.”(P 7)
“At this point, I felt like intrigued to see if I could get him to do it…could I get a horse to kick a ball, I was just hoping he would just sort of walk into it and kick it, so I was feeling like ripped at this point, as I had got the horse to kick a ball.”(P 6)
“There was a feeling of accomplishment there you can see it on my face the smiles and I felt like I did not do too badly, and at that point, I was thinking I am starting to enjoy the experience and it was becoming easier.”(P 3)
“I was really proud of myself.”(P 2)

The overarching emotional experience and emotional change expressed by the participants on completion of the activities were focused on positive feelings.
“Oh, I felt so happy, relaxed and calm, he came to me, I felt accomplished.”(P 2)
“I was inspired and encouraged, it feels amazing.”(P 7)
“He was funny, he made me happy.”(P 3)
“By the end, I felt noticeably different, I had a real sense of achievement and I felt much more relaxed than usual across my body, my back, and shoulders.”(P 6)
“I felt like proud and happy...no words, I need like a synonym...”(P 4)
“It really feels amazing.”(P 1)
“That made me happy with scoring a goal, like yeah, was happy…I felt accomplished and happy.”(P 5)
“You feel like you have achieved something, a new skill…you felt you had made a change. I still feel it now, though my body feels more relaxed than it does normally and in such a short amount of time, yeah, it was good.”(P 8)

The feelings of accomplishment and achievement were intense for the participants, and interestingly, one of them referred to being aware of relaxation in their body, another area which would benefit from further study in relation to this intervention.

## 13. Seeking and Building a Connection with the Horse

While undertaking interactions with the horse, participants were asked to build cooperation and a partnership with the horse. Seven of the participants very consistently expressed the experience of building a connection, and interestingly, a desire to build it from the start of their interaction with the horse.
“I felt tense and uncertain. The horse did not seem to be interested in me.”(P 2)
“I felt nervous and confused. The horse was not interested.”(P 3)
“It really worried me that he did not react to me. Like I knew he was calm, but when I was told to approach him and he did not do anything it worried me, like if I was doing anything wrong.”(P 5)
“Anxious really, the horse was ignoring me, indifferent. I was worrying the horse was not enjoying it.”(P 7)
“Awkward. I did not think the horses were really feeling anything.”(P 1)
“He did not really respond.”(P 6)
“He made me happy when he licked my hand.”(P 4)
Participants (5) also expressed a growing appreciation of the capacity of the horse in relation to communication and intelligence.
“I could not believe how they responded from the tiniest bit of energy.”(P 5)
“I felt like the horse was so clever, I had just pointed…made him go around in a semi- circle and then just like pointing…had corrected that and I thought that these horses are just so clever.”(P 4)
“It was really interesting about yeah how they read us so well.”(P 6)
“It feels really amazing, he’s listening to me waiting for me, I was speaking to him in my body and he was understanding.”(P 7)
“I was not sure. I had not been around horses, I didn’t know they could do things like that.”(P 3)

Again, the participants referred to being surprised by how clever and responsive the horses were.

All participants referred to the “connection” with the horse evolving as time went by.
“When you’re first looking at them it’s like a first date, your eyeing them up…and then it feels like you are getting somewhere and he just sniffed my hand, which is like well your welcome to continue and I think I tried to pet him and he was like no…like me going too fast too early.”(P 1)
“I wiggled my finger and he moved without saying anything, there has been no verbal communication, something is happening some kind of response.”(P 5)
“It was really nice cos (the horse) was nuzzling me and it was just, you know, you just keep on developing this relationship…it’s emotions, it’s trust, it’s responsibility, it felt like an approval, just like once again when he smelled my hand…then to like him nuzzling me. I could feel like there was this bond being created.”(P 3)
“By the end, I did feel a lot more connected, it was a nice feeling…knowing it’s a quick animal, it’s so relaxed and calm and it’s so easy to control as well. It was quite weird as I had not done that much with animals, so knowing I had got that connection was nice it made me happy. It was an enjoyable experience.”(P 4)
“It felt like we had really achieved something from the beginning to the end on a sort of journey, as we did not know each other and we sort of got to know each other and then we were together and developing my abilities…I felt like it was a journey between us and we had actually created a bond together.”(P 6)
“So, it felt like he responded a lot more cos I put a lot more like energy into it. I put my arm up like he was more aware of me and like the instructions were more clear.”(P 7)
“I felt like we were the team at the end….like well done.”(P 9)

Feeling a bond and like a team with the horse was expressed consistently by the participants showing the ability of the horse through this intervention to elicit strong positive emotions from human beings, as well as fear/nervousness as articulated at the start of the intervention. This range of emotion may be different from that expressed when interacting with other animals used in educational or therapeutic interventions such as dogs. However, this needs further study as articulated at the end of this paper in relation to future research plans.

## 14. Awareness of My Body

The participants (6) expressed an increasing awareness of their own body directly linked to this growing feeling of connection with the horse. This seemed to be related to communicating with the horse to get the tasks done through an embodied route.
“I started to assess my body and see how I was actually feeling in the moment in relation to the horses and how the horses are and the effect they are having on me…how I respond to them came a bit later but I was thinking how am I feeling and I was becoming a bit more aware of my body and what was going on.”(P 3)
“It takes it from just being this physical menial task to being something much deeper and much more psychological and emotional and I think that was then I started to gather more of an understanding…I was trying to be loose and trying to have some fun with myself cos I knew that the horses would react better if I was more playful…this is energy coming in and going out.”(P 2)
“In the end I felt a sense of achievement and more relaxed than usual.”(P 6)
“I was so not like expecting it to happen I was just like standing there and pointing and he was doing everything I had relaxed so much.”(P 7)
“Felt more like a unit…horse accepted me, we both felt more relaxed.”(P 5)
“I wanted to stroke him again, it was the contact…communication with a living being, it’s like another living being…it’s like you want to be with someone whose very honest and true and natural.”(P 9)

Referring to feelings of relaxation as the intervention progressed was consistent with the outcomes collected in relation to this intervention from those who refer individuals currently [30], indeed, the strongest outcome recorded two months after the intervention for participants is the ability to be calm.

## 15. Connections

The sense of growing a conscious connection with the horse seemed to be directly related to the positive learning experience for participants. The participants appeared to increase their understanding of themselves, which seemed to enable them to build a connection and achieve success with the horse. 

The emotions expressed on completing their interaction with the horse were universally very positive and linked to the connection or bond they felt with the horse.
“It felt really natural cos everything had washed away, and I was completely calm and I thought to myself this is crazy, at the start I did not even want to approach the horse and now I am just happily stood here stroking the horse.”(P 1)
“Contented and calm and happy…proud, I loved it, being with the horse.”(P 3)
“Being able to connect with the horse made me happy…it was a positive thing for me I felt happy and intrigued.”(P 5)
“Made me feel like I had achieved something and me and the horse had a bond together as he was listening to me and performing the actions I asked him to do, together, like I was the leader of the situation.”(P 6)
“I really developed as a person doing it with the horse.”(P 7)
“I felt accomplished and happy that the horse responded positively to me.”(P 2)
“I think with every little thing you do right it builds your confidence (the) more you gain a new skill. I felt like more entwined with the horses.”(P 8)

These findings agree with a previous study which found feelings of pride in young people engaging in interactions with horses [29]. The words used by participants were not in any way prompted by interactions with the research team who just asked them to report any feelings or emotions or changes in feelings and emotions in their mind or body before they started to watch the video. As university students, this group were articulate about their feelings and able to express them. 

It appeared from our analysis of the interview findings alongside the video recordings and the psycho-physiological data that participants were not specifically aware of the moments when the psycho-physiological arousal was recorded through the SCR. In some participants, they expressed being puzzled or confused in relation to some of these “events’’; however, there were equally as many recorded psycho-physiological “events” when they expressed no comment on their feelings or emotions when watching themselves on video. No pattern appeared in the analysis for this small group of participants, and further study with larger groups is needed to further understand whether participants are cognitively aware of any emotional arousal and when. There appeared to be a cyclical nature to the emotions experienced while learning the individual tasks which was repeated for each task and then the learning experience as a whole moving from confusion, to understanding and success, building a connection with the horse and creating an overall positive mood by the end.

## 16. Psycho-Physiological Data Results

The SCR response is a phenomenon where the skin momentarily becomes a better conductor of electricity when either external or internal stimuli occur that are physiologically arousing. Arousal is a broad term referring to overall activation and is widely considered to be one of the two main dimensions of an emotional response. Measuring arousal is, therefore, not the same as measuring emotion, but is an important component of it. Arousal has been found to be a predictor of attention and memory as discussed earlier in the paper. The stimuli to which skin conductance is sensitive are manifold, including in novel, significant or intense situations or experiences. Many different kinds of events can elevate your response including strong emotion, a startling event or a demanding or novel task.

To date, the research team has reprogrammed hardware in order to enable extraction of the raw data required for this study, and then Dartfish™ software was used, which allowed the creation of descriptive buttons, that when pressed “tag” the video with a time marker. Videos were recorded from a static camera with a wireless microphone. Four researchers considered the psycho-physiological data to initially identify where consistent SCR responses were occurring, and all researchers agreed on when consistent responses were occurring within the SCR recordings. These responses were then synchronised with the interview and video data in order to further illuminate the participants’ awareness of what was happening emotionally at that point (through the interviews) and the learning events which resulted in psycho-physiological responses (through the video data), respectively. Findings were then cross-matched once again to the psycho-physiology SCR findings to check the timing of events across the three datasets in order to ensure rigour in the analytical process. All observations were undertaken by a minimum of four academics, with most undertaken by all five members of the team.

The psycho-physiology results suggest that the participants’ skin conductance response (SCR) was activated by learning to ask the horse to do something. The following figures show our data on all nine participants and their SCR changes occurring throughout the learning experience (Figure 1); a still from the video footage of what the participants were doing when the “more developed manoeuvres” readings were taken is also presented (Figure 2). The skin conductivity response is presented here as “electro dermal activity” (EDA) presented in microSiemens (μS) on the graph showing the range of electrical response measured on the *y*-axis and the time along the bottom of the graph to show where in the participants’ learning experience the responses occurred on the *x*-axis. These responses were also divided into three sections: when the participants were introduced to the horse; what they would be doing (introduction) when the participants were learning how to play the games with the horse being directly managed by the instructor at this point (basic movements with horse and trainer); and when the participants were undertaking more advanced manoeuvres managing the horse on a long rope (more advanced manoeuvres with the horse and no trainer). 

In Figure 2, we have shared example illustrations of three manoeuvres all participants undertook in a still photograph to offer further insight into what the participants and the horses were doing when the participant was undertaking more advanced manoeuvres managing the horse on a long rope (more advanced manoeuvres with the horse and no trainer are shown in Figure 1).

## 17. Discussion

Interestingly, in the interviews while the participants watched themselves on video, they were not consistently aware of their own embodied emotional arousal; however, all participants stated that in the second half of their play with the horses (fifteen to twenty minutes into the data collection period), they started to express feelings of calmness, happiness, and achievement. This concurs with our initial ideas about this learning, that it is occurring initially through our emotions and our bodies rather than cognitively, and our pilot findings suggest that cognitive recognition of changes in mood or emotion seemed to happen later although obviously further research is required to specify timings in relation to these changes with a larger sample of participants. The research has demonstrated interesting insights into the emotional journey of undertaking elements of the EAI. For these students, these emotional changes were achieved in 35 minute periods of being taught elements of the course, and being able to rehearse them with the horses and then watch themselves on video. 

The EAI under study here has been shown through independent evaluation to have positive impacts and is currently being further researched through a feasibility study with 155 young people between the ages of 8–18 for whom “talking” interventions have failed. The course has already been shown in a small sample to have a potentially positive impact on one-year reoffending rates in young men incarcerated in a young offender’s institution who also showed some positive changes in problem behaviour following the intervention [2]. This charity has a clearly designed intervention which lends itself to study through the type and clarity of horsemanship [47] taught and the consistency of coaching and facilitation undertaken. 

It is also important to consider that it would appear from emerging literature on this area that the nature of the horse as a prey animal and the skills required to successfully interact and learn with a prey animal may be the characteristic which helps humans to control their behaviour, and in this case, learn to be calm [36]. Although of course further research is required to see whether, for instance, dogs can achieve similar results in the humans they interact with. 

The initial emotions expressed by our participants were dominated by anxiety, fear and nervousness, and all of them reflected extensively on the imposing presence of this large, charismatic, unpredictable mammal. We suggest that these emotions may focus the participants’ embodied responses on finding a way to successfully communicate and build rapport with this other “being”. As the participants rehearse the embodied skills required to enable them to “play” with the horse and ask them to do things and become successful, the “learning” is rehearsed and reinforced through their bodies, enabling them to feel, in the words of participants, “powerful”, “like a leader” and “calm”. Arguably, this embodied pre-speech capability to communicate with each other as with young children, and as in this case, communicate and learn with a non-human animal [64] using “inter-natural” embodied emotional mechanisms may offer an opportunity to interrupt previous potentially maladaptive habits developed to deal with emotional arousal [65]. This learning process may indeed be offering a safe opportunity to rehearse a successful calm outcome following emotional arousal.

We contend that natural horsemanship may be operating at an embodied, emotional level, that is, that the learning could be embodied, and emotion led to aid learning and behaviour change. The young people responded emotionally to the presence and responses of the horse, and as they are coached to success repeatedly, that positive emotional landscape was rehearsed and reproduced from 100–200 times on average for an individual referred to the course (calculated estimate based on the participants thirty five minute experiences). These positive embodied emotional responses then generalise to everyday life, through dealing with emotional arousal as it occurs, thereby creating the mechanism of action. 

The people who normally attend this course have a variety of different behavioural and mental health issues, and as outlined earlier, many of which are associated with abuse and neglect or witnessing violence in the home. Interestingly, through evaluation of the course, they all leave with similar benefits, the most consistent and strong of which is calmness in our evaluations so far, thus enabling participants to re-engage with education and relationships, with those who refer to the course (primarily social workers) reporting rapid and effective changes in behaviour, as one participant said following completion of the course: “I feel reborn”. Does using embodied emotion-led interventions offer the opportunity for those for whom “talk”-based interventions are not working to gain therapeutic benefit? 

A recent article [66] focusing on the effects of child abuse and neglect would suggest that both yoga and mindfulness are ways of successfully helping affected individuals and the intervention under study here has been referred to as “mindfulness in action”. Offering as it does a practical way to rehearse being in the moment, both calm and present, which is the only effective way to learn to communicate with the horses successfully.

The results from this study may be helpful across areas of practice both professionally and academically, such as practice-based learning with animals and without, mental health and social care interventions and even mainstream education in relation to providing opportunities to engage with all types of learners. The methodology may be of interest to many disciplines such as psychology, psychiatry, nursing, public health, social work, etc., offering the opportunity to examine the emotional state and lived experiences of participants in learning interventions designed to improve health and wellbeing and influence behavioural change. 

In addition, this intervention currently works with individuals aged 8–80 although between 60–65% of referrals in any one year are for under 18-year-olds. In future studies, it would be important to consider developmental stages in relation to the age range of participants and the actual and potential outcomes that could be achieved. The study presented here, however, used a group of 18–24-year-olds as it was felt to be unethical to expose the vulnerable participants normally referred to the intervention to this type of pilot research involving exploration of emotion without a clear justification in relation to the potential insights we may have gained from the study. 

In many ways this pilot study has raised more questions than it has answered. Does learning through this mechanism enhance attempts to change behaviour? Does this mechanism only occur in challenging learning situations, such as in this case, learning to communicate across species with large charismatic and unpredictable prey animals? Would the learning mechanism be the same across other species such as humans and dogs? Further studies are needed to understand this mechanism, and exactly when it occurs and its potential benefits.

## 18. Limitations

It may be of course that our participants were aware of “feelings” related to the recorded emotional arousal occurring at certain points while they undertook the course, but they may have not had the “language” to describe this or may have been unwilling to share these feelings in the interviews. Watching themselves on video may not be an adequate way of “reliving” the experience in order to recreate or re-stimulate the emotional learning psycho-physiological responses effectively or accurately. As we only undertook recording these responses in real time with two out of the sample of participants, further exploration is required. All of the participants were university students, therefore their desire to be “successful and not fail” may have influenced the findings.

## 19. Further Research 

Within proposed future study on this area, the researchers would further refine these descriptors of significant psycho-physiological events in the EAI and enhance the reliability using blind-inter rater reviewing. For the purposes of this study, the researchers undertook the analysis as a group together rather than separately. In addition, a four-group experimental design would be used in order to attempt to understand whether emotional arousal occurs, for instance, with other non-human animals used in animal-assisted interventions, such as dogs. Through this process, we will refine our understanding of significant learning events, emotional arousal, human–animal interactions, and behavioural change.

## Figures and Tables

**Figure 1 animals-09-00303-f001:**
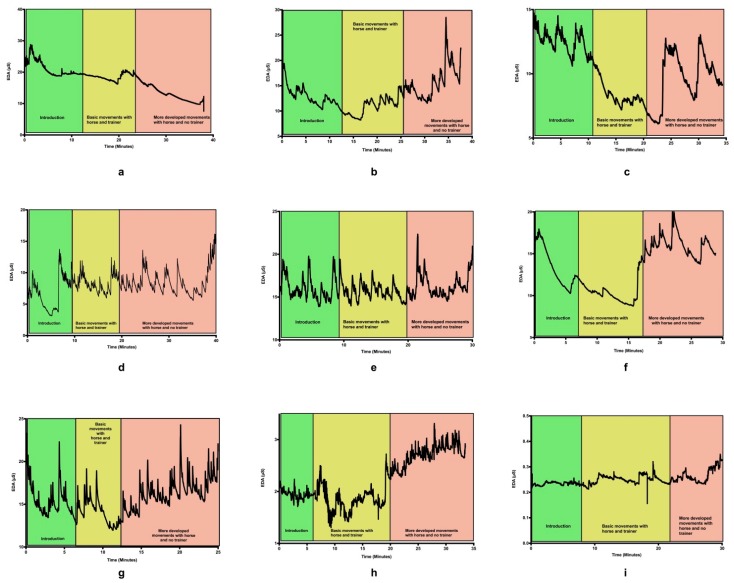
(**a**) Participant 1 SCR (skin conductivity responses), (**b**) Participant SCR, (**c**) Participant 3 SCR, (**d**) Participant 4 SCR, (**e**) Participant 5 SCR, (**f**) Participant 6 SCR, (**g**) Participant 7 SCR, (**h**) Participant 8 SCR, (**i**) Participant 9 SCR.

**Figure 2 animals-09-00303-f002:**
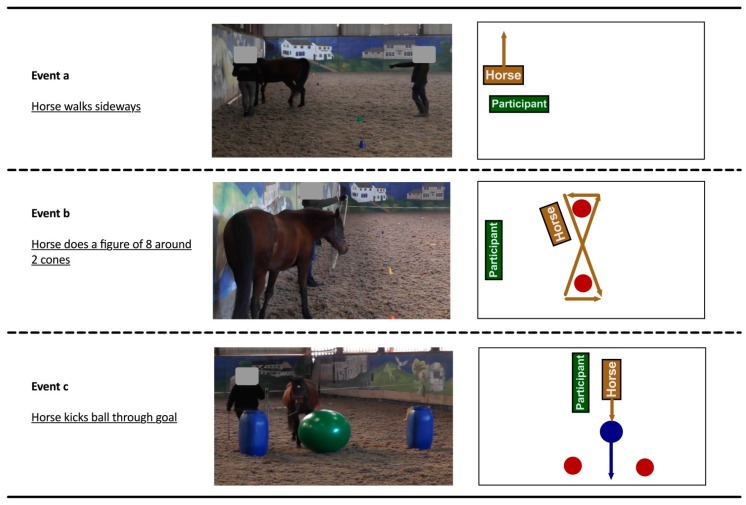
Examples of more advanced manoeuvres with horse under the direction of the participant.

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
