# Peer review of "An Exploration of the Mechanism of Action of an Equine-Assisted Intervention"

_animals, 2019, doi:10.3390/ani9060303_

Round 1

Reviewer 1 Report

This is a well written paper based on an interesting and informative study. There are a few areas that I think need further clarification:

I found the first 3 paragraphs of the 'abstract' more straightforward, informative and eaiser to understand than the 'simple summary' so wonder if some of the same langauge could be used in the latter to be more informative to the general reader

Methods -  How as it all analysed and compared? Can you triangulate these very different kinds of data? How? Whilst discussion of ethics approval is included, how did the rseaerch team try and put the young people at ease and account for thier identified behavioural needs in the study? 

Presentation of results and discussion - it was not clear to me how the two main 'types' of data worked together - the interviews and psycho-physiological results. Could this be made more explicit? If they were 'traingulated' as suggested, what did this reveal? The presentation of the interview data also lacks development - it is very much just 'presented' and I think more narrative around it would help draw out the interesting findings more clearly.

Human-animal interactions - I was surprised there was little discussion of human-animal interactions and related litertaure, as this study seems to be all about this. Does the interspecies aspect of the enounter matter? Does it matters it's with horses, rather than say dogs?

Author Response

Response to reviewer comments on the paper - An Exploration of the Mechanism of Action of an Equine Assisted Intervention 15/5/19 Ann Hemingway

Reviewer 1

Many thanks for your useful comments I have replied to them here in red and in the document using track changes and inserting comments to highlight where your specific changes have been made. Yours Ann Hemingway

Comment 1 - I found the first 3 paragraphs of the 'abstract' more straightforward, informative and easier to understand than the 'simple summary' so wonder if some of the same language could be used in the latter to be more informative to the general reader.

Many thanks I have made changes the abstract and simple summary as you suggest on pages 1 and 2 of the document lines 6 to 57.

Comment 2 - Methods - How as it all analysed and compared? Can you triangulate these very different kinds of data? How? Whilst discussion of ethics approval is included, how did the research team try and put the young people at ease and account for their identified behavioural needs in the study? 

Many thanks I have responded to your comments on pages 7 and 8 lines 298 to 302 on page 7 and lines 341 on page 7 to 345 on page 8.

Comment 3 - Presentation of results and discussion - it was not clear to me how the two main 'types' of data worked together - the interviews and psycho-physiological results. Could this be made more explicit? If they were 'triangulated' as suggested, what did this reveal? The presentation of the interview data also lacks development - it is very much just 'presented' and I think more narrative around it would help draw out the interesting findings more clearly.

Many thanks I have responded to your comments on the following pages, page 8 lines 366 to 368, Page 10 lines 473 to 478, page 11 lines 496 to 503, page 11 lines 529 to 531, page 12 lines 552 to 554, page 13 lines 588 to 590, page 14 line 618, page 14 lines 642 to 647, page 15 lines 669 to 672, page 15 and 16 lines 697 to 700, page 16 lines 709 and 710. Also pages 18 and 19 lines 786 to 797 and page 16 lines 703 to 713.

Comment 4 - Human-animal interactions - I was surprised there was little discussion of human-animal interactions and related literature, as this study seems to be all about this. Does the inter-species aspect of the encounter matter? Does it matters it's with horses, rather than say dogs?

Many thanks this is an important point and mentioned by another reviewer I have attempted to address this at several points in the paper. Page 3 lines 119 to 126. Page 11 lines 496 to 503. Page 14 lines 642 to 647. Page 20 lines 873 and 874.

Reviewer 2 Report

Brief summary

This paper presents a pilot study investigating a possible mechanism of action in equine assisted therapy. The authors developed a new methodology for investigating the mechanism of action including three data collections measures: video data, qualitative interview data, and psycho physiological data (SCR). Study participants included nine healthy young adults who, for the purposes of the study, followed part of an established EAI program that has been used with over 500 people. Results shed light into the positive outcomes associated with this specific program and animal assisted therapy in general. Specifically, the program intervention appeared to stimulate emotional arousal and increased calmness and positive mood (originating from, for example, feelings of pride and connection with the horse).

Broad comments

This is an innovative study that, importantly, addresses a possible mechanism of action in animal assisted interventions. As the authors point out in their introduction, studies on animal assisted therapy often lack rigor and show mixed results on positive outcomes. Many of the studies to date set out to establish if animal assisted therapy leads to positive outcomes and not how they might do so. This study, examining a possible mechanism related to behavioral outcomes due to an EAI, makes an important contribution. In particular, I thought it was a novel and interesting approach to study a large-scale and successful program with a separate group of healthy participants instead of participants actually participating in the program for therapeutic reasons. Another strength of the study was the inclusion of three different data sets.

Specific comments 

1.     What ‘evaluation’ are the authors referring to in line 180? Do they mean as established by the Parelli Natural Horsemanship program?

2.     A typo for the word “study” on line 306 and “learners” on line 681.

3.     The sentence, “The words emotions, feelings and sensations were used in an attempt…” starting on line 354 and ending on line 358 is confusing. In the previous sentence it is explained that the participants were asked to share their experience of any “emotions or feelings”. So, the addition of the word ‘sensation’, or perhaps just the long length of the following sentence does not completely make sense.

4.     A bit more detail or clarification is needed in the Psycho Physiological Data Results section – specifically related to the sentence on lines 586-587. The sentence states that four researchers identified where consistent SCR responses were occurring. What criteria did they use? Were they always in agreement? It is also unclear what is meant by “the findings were then cross matched once again…” (line 590) or why four or all five researchers were involved in the observation analysis. In the Further Research section it is mentioned that blind-inter rater reviewing could enhance reliability (line 702), but it is unclear what the current reliability was.

5.     The wording “by referrers” (line 675) is confusing (and does not seem necessary). Similarly, “learning” on line 678 would make more sense as “results”.

6.     Inconsistent punctuation of article titles in the reference list.

Author Response

Response to reviewer comments on the paper - An Exploration of the Mechanism of Action of an Equine Assisted Intervention 15/5/19 Ann Hemingway

Please note these changes are also highlighted as comments in the revised text of the article.

Reviewer 2

1.     What ‘evaluation’ are the authors referring to in line 180? Do they mean as established by the Parelli Natural Horsemanship program?

Many thanks this has been further explained on page 5 line 244.

2.     A typo for the word “study” on line 306 and “learners” on line 681.

Many thanks these have been corrected page 9 line 399 and page 16 line 743.

3.     The sentence, “The words emotions, feelings and sensations were used in an attempt…” starting on line 354 and ending on line 358 is confusing. In the previous sentence it is explained that the participants were asked to share their experience of any “emotions or feelings”. So, the addition of the word ‘sensation’, or perhaps just the long length of the following sentence does not completely make sense.

Many thanks this has been further explained in the text on page 10 lines 443 to 453. In addition on pages 18 and 19 lines 786 to 797.

4.     A bit more detail or clarification is needed in the Psycho Physiological Data Results section – specifically related to the sentence on lines 586-587. The sentence states that four researchers identified where consistent SCR responses were occurring. What criteria did they use? Were they always in agreement? It is also unclear what is meant by “the findings were then cross matched once again…” (line 590) or why four or all five researchers were involved in the observation analysis. In the Further Research section it is mentioned that blind-inter rater reviewing could enhance reliability (line 702), but it is unclear what the current reliability was.

Many thanks this has been further explained on page 16 lines 728 to 739.

5.     The wording “by referrers” (line 675) is confusing (and does not seem necessary). Similarly, “learning” on line 678 would make more sense as “results”.

Many thanks these have been corrected please see page 20 line 849 and page 20 line 853.

6.     Inconsistent punctuation of article titles in the reference list.

Many thanks these have been corrected throughout the reference list.

Reviewer 3 Report

This manuscript is based on examining the in-situ responses of 9 individuals exposed to engagement in equine assisted therapy. The main aim of this manuscript was to examine potential mediating processes that may underlie significant effects of equine assisted therapy approaches on various outcomes. ‘Potential mediating variables’ are examined using several methods to capture aspects of emotion or emotion states, emotion regulation and/or arousal. Emotion - based learning and arousal were measured through simultaneously examining unstructured, qualitative, subjective perceptions and descriptions of participant-monitored emotion states through interview data and skin conductance.

While I am excited about the aim of the study – understanding the underlying mechanisms that underlie effects of equine assisted therapy on participants functioning in several domains, there are several issues with the rationale, research design and methods, and reporting of results that limit my enthusiasm about this manuscript.

Introduction: First of all, I would suggest that it is common in any field to pursue what is often referred to as first-generation research – identifying the efficacy of a program or intervention. This is done by conducting efficacy trials which are based on using causal approaches i.e. randomly assigning participants to treatment (comparison) conditions, including pre and posttest assessment and examining whether difference hypotheses can be confirmed. It is then, and only once we know causal impacts are present, that we start to be fully justified in examining pathway analyses – conducting what is at times referred to a second generation research in a field. In particular, we need to first identify and summarize causal work conducted in the field of equine assisted therapy to infer that there are causal effects ( and ideally do so by examining for whom and under what condition? And on which outcomes? ) before we start to examine why this may be the case – and ask :”what is the underlying mechanism?”

While there certainly is some causal work in the equine literature, I would argue that causal designs are severely lacking in the field and it is not at all clear that efficacy has been demonstrated. Certainly, a discussion about various types of approaches in subfields need to be addressed first – therapy? Equine assisted learning? Hippotherapy? Therapeutic riding? What do we know about these particular kinds of interventions for which types of populations and under what conditions? Which outcomes appear to be most affected? That overview is not provided by the authors. Instead they have summarized ‘the field’ making a lot of causal statements. Research listed from 65-83 are all referring to causal language and this is simply not accurate. In addition you can’t say, if you mention that reviews lacked ‘lacked rigour, randomisation and adequate sample size’ it makes no sense to follow that by suggesting that ‘ however those studies that have been published do report positive outcomes across a range of physical, behavioural and social areas.

As a reader who knows the field, the descriptions of existing work creates a concern. In addition, the review of causal work is incomplete, which is unfortunate. It sets up an impression with the reader that the authors are unaware of the difference between causal and correlations designs and have a naïve view of the state of the field, which makes one question the ensuing sections- I am sorry as I don’t mean to be harsh but this is a problem.  In sum, the authors have not summarized the findings fully or accurately, which in my opinions somewhat limits the case that should provide the rationale for their approach in examining causal mechanisms.

In general, it is important to more clearly describe and define what makes your intervention ‘therapy’.

I would suggest that authors refer to the notion that the definitions of the different types of equine approaches are by no means agreed upon in the field and differ when discussed from a veterinary angle, a mental health perspective or horsemanship source. I think it is important authors refer to definitions used by PATH international which have some specific comments about what constitutes therapy which refers more the education and professional background of practitioners and the population served, and the goals outlined, than the nature of the activities themselves. Some of the other approaches are not mentioned.

Moreover, there needs to be a clearer description of the role of the therapist in this as it is the therapist who is centrally charged with identifying the links between the behavior of the horse and the clientand helping the clients think and feel differently about the events on hand in ways that shape the client’s behavior which in turn shapes the behavior of the equinewhich also need to be contextualized during these interactions.

The overview of topics and ‘games’ can be supported intuitively, they are less compelling as a stand-alone feature unless a strong rationale is provided that exemplifies use of equines as facilitating the application of these principles and their link to emotion. Unless these aforementioned connections are made explicitly, it is necessary to describe each weekly lesson activity and describe how it taps into the underlying role of emotion.

On a similar note, there is an emerging literature providing several models identifying how characteristics of equines inform human equine interactions in ways that require participants to regulate their emotion (fear), behavior and cognitions, the existing manuscript could be much strengthened by describing what the unique features of equine approaches are and how they tie into the proposed framework. The listed activities could be used to illustrate this through examples. While the authors refer to these links, they are not clear to the average reader; I believe the manuscript would be stronger by unpacking this in significantly greater detail.

What is it about the equine size? Prey animal? Sensitivity to the herd, hierarchy? - that makes them behave in ways that shape the behavior of clients in ways that require   self-regulation, awareness of emotionfear and guides their cognitions – with help of the therapist - to overcome those fears in ways that non-verbally communicate intent - and leadership– to the equine? How does the equine respond – at what point – and what is the role of pressure or intent or cognitions or subtle changes in body langue of the clients that lead the equine to respond in preferable, predictable ways that teach the client the power of their cognitions shaping their emotion and behavior – to shape equine responses?

Next, given that authors are referring to children and youth and adults, you need to identify the ways in which age i.e., development may come into play. For example, abstract thinking, evaluating one’s own emotion and meta-cognitions that play a role in guiding emotion are not developmental milestones reached until adolescence and for some not until they are well underway into adolescence. The authors need to attend to the issue of development… how do these approaches apply to populations of different ages?

This could be discussed in the discussion.

In sum, there is a lot of lead in in the introduction and spillover in the methods about the role of emotion and learning that can be reduced substantially to focus on the direct links between the nature of the intervention, existing models that have been postulated about underling effects, and a thorough review of a very small body of causal work to highlight why you are looking as various aspects of emotion as your main suspect for mediation.

Method and results: There are several aspects that represent in my opinion unconventional approaches towards sharing the methods used.

There needs to be a more traditional description of recruitment and screening method for your study. Authors describe using ‘Nine healthy participants’ – but fail to conclude participation variable descriptions and common indices about how these participants were sampled and recruited/compensated? Also, what do you mean ‘healthy’ individuals…? You previously referred to the intervention being targeted at ‘young people with chronic mental health and 30 behavioral problems for who talk based interventions are not working’ – what is the relationship between your sample and the population this intervention is commonly targeted at?

Listing this ‘form’ on line 192 – 210 is not productive way or APA style nor is it usually described in this way in a sample description

When there are only 9 participants, speaking of a cohorts seems odd

I am not sure referring to the fact that 500 individuals have undertaken this intervention is contextualized enough. I don’t think it belongs in this section.  In my opinion, it is exactly why we should be concerned; we are implementing and expanding interventions before efficacy trials have been conducted. In addition, I am sorry to say that it set up the expectation about sample size – to find out later that your N = 9 is a little bit of a letdown.

The term ‘exploratory research’ study doesn’t really refer to the fact that it combines both physiological and experiential methodologies; exploratory is something different. Also the word choice of 3 data sets is odd as well/you are simply measuring 3 different methods to capture components of the same construct – emotion.

The measures are severely limited in their description. Authors describe that participants “were asked to stop the video and share any changes in emotion at any point while watching – this methodology needs to be described in details. How were they asked to rate their emotion? Using a reliable, tested psychometrically sounds toll? Open-ended questions? How were these coded? Who decided the criteria for stopping the video? Based on what types of behavior?

How did you code these interview data?

As you describe your measures, you are better off keeping the descriptions and rationales using terms such as Hetero phenomenology/Neuro phenomenology to a minimum as a regular developmental audience is going to refer to this as emotion states, or affect, possibly aspects of emotion regulation. That whole rationale can be reduced.

Authors say: The following methods of data collection were used to inform this study: - this needs descriptions of how. How did you measure this? At what time points? How were participants instructed? Especially the skin conductance portion needs explanation for the uninitiated. Then you need to take the reader through how these variables were transformed into something that we can interpret.

Results: You need to describe your analytic strategy and report results using conventional approaches or descriptions.

Some general comments:

In several instanced hyphens are need e.g., talk-based, disability-related issues, emotion-based learning

 Abstract: I would suggest being more direct and to the point about the purpose of the study: introduce the idea of examine mediating pathways that may underlie effects of equine assisted intervention and that you are examining the role of emotion

 The intervention under study is ‘used’ – implemented would be a better word

Authors refer to ‘primary learning process in the intervention was an embodied emotional response. – what does this mean?

Author Response

Response to reviewer comments on the paper - An Exploration of the Mechanism of Action of an Equine Assisted Intervention shown in red here 15/5/19 Ann Hemingway

Please note these changes are also highlighted as comments in the revised text of the article. Many thanks for your clear and constructive comments.

Reviewer 3

General comment 1 - This manuscript is based on examining the in-situ responses of 9 individuals exposed to engagement in equine assisted therapy. The main aim of this manuscript was to examine potential mediating processes that may underlie significant effects of equine assisted therapy approaches on various outcomes. ‘Potential mediating variables’ are examined using several methods to capture aspects of emotion or emotion states, emotion regulation and/or arousal. Emotion - based learning and arousal were measured through simultaneously examining unstructured, qualitative, subjective perceptions and descriptions of participant-monitored emotion states through interview data and skin conductance.

General comment 2 - While I am excited about the aim of the study – understanding the underlying mechanisms that underlie effects of equine assisted therapy on participants functioning in several domains, there are several issues with the rationale, research design and methods, and reporting of results that limit my enthusiasm about this manuscript.

Responses to all specific comments are outlined below in relation to each specific point many thanks.

3.    Introduction: First of all, I would suggest that it is common in any field to pursue what is often referred to as first-generation research – identifying the efficacy of a program or intervention. This is done by conducting efficacy trials which are based on using causal approaches i.e. randomly assigning participants to treatment (comparison) conditions, including pre and posttest assessment and examining whether difference hypotheses can be confirmed. It is then, and only once we know causal impacts are present, that we start to be fully justified in examining pathway analyses – conducting what is at times referred to a second generation research in a field. In particular, we need to first identify and summarize causal work conducted in the field of equine assisted therapy to infer that there are causal effects ( and ideally do so by examining for whom and under what condition? And on which outcomes? ) before we start to examine why this may be the case – and ask :”what is the underlying mechanism?”

Many thanks these comments have been addressed in the following places in the paper page 1 line 7, page 2 lines 63-65, page 3 lines 3-5, and lines 19-26.

4.    While there certainly is some causal work in the equine literature, I would argue that causal designs are severely lacking in the field and it is not at all clear that efficacy has been demonstrated. Certainly, a discussion about various types of approaches in subfields need to be addressed first – therapy? Equine assisted learning? Hippotherapy? Therapeutic riding? What do we know about these particular kinds of interventions for which types of populations and under what conditions? Which outcomes appear to be most affected? That overview is not provided by the authors. Instead they have summarized ‘the field’ making a lot of causal statements. Research listed from 65-83 are all referring to causal language and this is simply not accurate. In addition you can’t say, if you mention that reviews lacked ‘lacked rigour, randomisation and adequate sample size’ it makes no sense to follow that by suggesting that ‘ however those studies that have been published do report positive outcomes across a range of physical, behavioural and social areas.

Many thanks responses to these points have been included at the following points in the redrafted paper. Pages 3 and 4 lines 103-104 and109-148.

5.    As a reader who knows the field, the descriptions of existing work creates a concern. In addition, the review of causal work is incomplete, which is unfortunate. It sets up an impression with the reader that the authors are unaware of the difference between causal and correlations designs and have a naïve view of the state of the field, which makes one question the ensuing sections- I am sorry as I don’t mean to be harsh but this is a problem.  In sum, the authors have not summarized the findings fully or accurately, which in my opinions somewhat limits the case that should provide the rationale for their approach in examining causal mechanisms.

Many thanks responses to these points have been included at the following points in the redrafted paper. Pages 3 and 4 lines 109-148.

6.    In general, it is important to more clearly describe and define what makes your intervention ‘therapy’.

Many thanks please find the response at page 3 lines 140-142.

7.    I would suggest that authors refer to the notion that the definitions of the different types of equine approaches are by no means agreed upon in the field and differ when discussed from a veterinary angle, a mental healthperspective or horsemanship source. I think it is important authors refer to definitions used by PATH international which have some 

specific comments about what constitutes therapy which refers more                           to the education and professional background of practitioners and the population served, and the goals outlined, than the nature of the activities themselves. Some of the other approaches are not mentioned.

Many thanks please find the response at page 3 lines 137-142. 

8.    Moreover, there needs to be a clearer description of the role of the therapist in this as it is the therapist who is centrally charged with identifying the links between the behavior of the horse and the client – and helping the clients think and feel differently about the events on hand in ways that shape the client’s behavior which in turn shapes the behavior of the equine – which also need to be contextualized during these interactions.

Many thanks please find the response at page 3 lines 137-142.  

9.    The overview of topics and ‘games’ can be supported intuitively, they are less compelling as a stand-alone feature unless a strong rationale is provided that exemplifies use of equines as facilitating the application of these principles and their link to emotion. Unless these aforementioned connections are made explicitly, it is necessary to describe each weekly lesson activity and describe how it taps into the underlying role of emotion.

Many thanks please find the response at page 6 lines 249 to 255. 

10.  On a similar note, there is an emerging literature providing several models identifying how characteristics of equines inform human equine interactions in ways that require participants to regulate their emotion (fear), behaviour and cognitions, the existing manuscript could be much strengthened by describing what the unique features of equine approaches are and how they tie into the proposed framework. The listed activities could be used to illustrate this through examples. While the authors refer to these links, they are not clear to the average reader; I believe the manuscript would be stronger by unpacking this in significantly greater detail.

Many thanks please find the response at page 19 lines 808-812. 

11.  What is it about the equine – size? Prey animal? Sensitivity to the herd, hierarchy? - that makes them behave in ways that shape the behavior of clients in ways that require   self-regulation, awareness of emotion – fear –and guides their cognitions – with help of the therapist - to overcome those fears in ways that non-verbally communicate intent - and leadership– to the equine? How does the equine respond – at what point – and what is the role of pressure or intent or cognitions or subtle changes in body langue of the clients that lead the equine to respond in preferable, predictable ways that teach the client the power of their cognitions – shaping their emotionand behavior – to shape equine responses?

 Many thanks please find the response at page 19 lines 808-812. 

12.  Next, given that authors are referring to children and youth and adults, you need to identify the ways in which age i.e., development may come into play. For example, 

Abstract thinking, evaluating one’s own emotion and meta-cognitions that play a role in guiding emotion are not developmental milestones reached until adolescence and for some not until they are well underway into adolescence. The authors need to attend to the issue of development… how do these approaches apply to populations of different ages?

This could be discussed in the discussion.

 Many thanks please find the response at page 20 lines 861-867, page 20 lines 861-867.

13.  In sum, there is a lot of lead in in the introduction and spill over in the methods about the role of emotion and learning that can be reduced substantially to focus on the direct links between the nature of the intervention, existing models that have been postulated about underling effects, and a thorough review of a very small body of causal work to highlight why you are looking as various aspects of emotion as your main suspect for mediation.

 Many thanks please find the response at page 4 lines 168-172 and page 4 lines 175-178. Page 5 lines 208-214.

14.  Method and results: There are several aspects that represent in my opinion unconventional approaches towards sharing the methods used. There needs to be a more traditional description of recruitment and screening method for your study. Authors describe using ‘Nine healthy participants’ – but fail to conclude participation variable descriptions and common indices about how these participants were sampled and recruited/compensated? Also, what do you mean ‘healthy’ individuals…? You previously referred to the intervention being targeted at ‘young people with chronic mental health and 30 behavioral problems for who talk based interventions are not working’ – what is the relationship between your sample and the population this intervention is commonly targeted at?

 Many thanks please find the response at page 6 lines 248-255, page 7 lines 298 to 302.

15.  Listing this ‘form’ on line 192 – 210 is not productive way or APA style nor is it usually described in this way in a sample description

Many thanks please find the response at page 4 line 71.

16.  When there are only 9 participants, speaking of a cohorts seems odd

 Many thanks please find the responses at page 1 line 42, page 16 line 709, page 20 line 864.

17.  I am not sure referring to the fact that 500 individuals have undertaken this intervention is contextualized enough. I don’t think it belongs in this section.  In my opinion, it is exactly why we should be concerned; we are implementing and expanding interventions before efficacy trials have been conducted. In addition, I am sorry to say that it set up the expectation about sample size – to find out later that your N = 9 is a little bit of a let down.

 Many thanks please find the response at page 1 line 15 and 16.

18.  The term ‘exploratory research’ study doesn’t really refer to the fact that it combines both physiological and experiential methodologies; exploratory is something different. Also the word choice of 3 data sets is odd as well/you are simply measuring 3 different methods to capture components of the same construct – emotion.

 Many thanks please find the responses at page 1 line 31, page 2 line 61 and 78, page 5 line 222, page 7 line 299, and 310, page 9 line 408.

19.  The measures are severely limited in their description. Authors describe that participants “were asked to stop the video and share any changes in emotion at any point while watching – this methodology needs to be described in details. How were they asked to rate their emotion? Using a reliable, tested psychometrically sounds toll? Open-ended questions? How were these coded? Who decided the criteria for stopping the video? Based on what types of behavior?

 Many thanks please find the responses at page 10 lines 454-465.

20.  How did you code these interview data?

 Many thanks please find the responses at page 10 lines 454-465.

21.  As you describe your measures, you are better off keeping the descriptions and rationales using terms such as Hetero phenomenology/Neuro phenomenology to a minimum as a regular developmental audience is going to refer to this as emotion states, or affect, possibly aspects of emotion regulation. That whole rationale can be reduced.

Many thanks please find the responses at page 7 lines 306-314, page 8 lines 352-354 and 361-366.

22.  Authors say: The following methods of data collection were used to inform this study: - this needs descriptions of how. How did you measure this? At what time points? How were participants instructed? Especially the skin conductance portion needs explanation for the uninitiated. Then you need to take the reader through how these variables were transformed into something that we can interpret.

 Many thanks please find the responses at page 10 lines 454-465 and 472-478, page 16 lines 717-739, page 17 lines 760-763.

23.  Results: You need to describe your analytic strategy and report results using conventional approaches or descriptions.

 Many thanks please find the responses at page 16 lines 717-739 pages 16 and 17 lines 745 to 754 and lines 760-763.

Some general comments:

24.  In several instanced hyphens are need e.g., talk-based, disability-related issues, emotion-based learning

 Many thanks please find the responses at page 1 line 8, line 13, line 15 and line 30. 

25.  Abstract: I would suggest being more direct and to the point about the purpose of the study: introduce the idea of examine mediating pathways that may underlie effects of equine assisted intervention and that you are examining the role of emotion

 Many thanks please find the responses at page 1 lines 34-36 and page 2 lines 52-54.

26.  The intervention under study is ‘used’ – implemented would be a better word

 Many thanks please find the responses at page 1 line 12.

27.  Authors refer to ‘primary learning process in the intervention was an embodied emotional response. – what does this mean?

 Many thanks please find the responses at page 2 lines 52-54 and lines 63-65.

Round 2

Reviewer 1 Report

A small comment - in the methods section on p.5 the particpants are referrred to as 'boys' and 'girls' when they are adults - I think this should be changed.

Other than that, I think the authors have responded well to review comments and produced a clear account of this interesting study and the findings. 

Reviewer 2 Report

All suggested changes and comments addressed in revised manuscript. 

Reviewer 3 Report

na